# New Forms of Social Learning in Mediterranean Higher Engineering Education: Change Lab for Gender Equality Transformation, Methodology, Design Principles

**Anastasia Zabaniotou** [1,2]

[1]   Department of Chemical Engineering, School of Engineering, Aristotle University of Thessaloniki, 54124 Thessaloniki, Greece; azampani@auth.gr; Tel.: +30-6945-990-604

[2]   RMEI-Network of Mediterranean Schools of Engineering and Management, Ecole Centrale Marseille, Technopôle de Château-Gombert 38, 13013 Marseille, France

**Abstract:** Educating students to understand the dynamics of complex systems and acting with responsibility and equality in their professional/private life is pivotal. Implementing social changes in higher engineering education institutions is a challenge. This work is grounded empirically in the tailored practices of the gender equality Change Lab embedded in the network of Mediterranean engineering schools, which is a collective organizational integrity. We used action research and practical learning in our methodology. Design principles are provided, and methodological setup is included. We advocate that in order to mainstream gender equality, develop structures, and overcome some of the known limitations, we need to have conceptual clarity, well-targeted interventions, reflexivity, and empirical evidence. Moving from separate bureaucratic centrally-managed higher education institutions to interconnected networks that can organize self-assembling collaboration in the form of labs, with mutually beneficial partnerships contemplating social innovations, can challenge the melting of the traditional boundaries towards inclusive education. This can be done amidst university reforms conducive to such a transformation. Particular attention is paid to the role of HORIZON 2020 Taking a Reflexive Approach to Gender Equality for Institutional Transformation (TARGET) project in conceptualizing gender equality learning and system change in Mediterranean engineering schools.

**Keywords:** Change Lab; network; engineering; education; Mediterranean; gender equality

## 1. Introduction

The cultural and social development of a country is associated with the quality of the education system to create the future socio-ecosystem's learners [1]. Although, traditionally the role of higher engineering education is to offer knowledge and skills for innovation experience with full benefits of new technologies that underpin innovative products, the 21st century's unprecedented global challenges beyond climate change call for responsible research and innovation and inclusion in universities. Under the pressure of these global challenges, education is expecting to be a catalyst for social change achievement. Universities should assume a privileged position as key drivers of education for sustainable development to integrate diversity and gender equality [2], while engineering schools are expecting to play an important role in shaping the future of the world's society by generating new knowledge, developing appropriate competencies, and raising sustainability, inclusion, and ethical responsibility awareness. By defining engineering as the social practice of conceiving, designing, implementing, producing, and sustaining complex technological products, processes,

or systems [3], it can be argued that some leadership capability, embedded responsibility, and ethics are also required [4].

A transition from a focus on technical knowledge to inclusive engineering education requires integrated, interdisciplinary, and participatory approaches, for preparing men and women to work together on sustainable solutions and benefit entire societies [5]. A participatory approach for dealing with systems innovations is required, using desirable future visions, stakeholders' involvement, and society's project-based training [6].

The fast-paced innovation landscape requires collaboration between multiple institutional stakeholders in Living Laboratories (Living Labs), while the social dimension of education requires Social Laboratories (Social Labs) for social change learning.

Living Labs represent an approach to user-centric innovation environment by engaging users actively as contributors to creative and evaluative processes in innovation and development [7], relevant partners in real-life contexts, aiming to create sustainability values [8]. Living Labs are established at the boundaries between research, innovation, and policy [9].

Social Change Laboratories are transdisciplinary initiatives that develop and apply innovative approaches for scaled and systemic transformation processes, by actionable intervention approaches. Social Change Labs function as vehicles for systemic change by experimenting with social innovations, offering spaces for doing social experiments in a practical context where experts and stakeholders join together to initiate actions focused on tackling challenges without knowing exactly how to proceed [10].

The generic Change Laboratory method was developed in 1997 as a way to carry out developmental work research based on methodology for studying and developing work practices in collaboration between the researcher and the practitioner [11].

Despite almost two decades of Living and Social Labs' activity all over Europe, there are a lack of practices and research frameworks [8,12]. A theoretical and methodological gap continues to exist in terms of Living and Social Labs literature; empirical research of practical implementations is also limited [13].

*Aim, Objectives, and Innovative Aspects of the Study*

This paper aims to introduce a dialogue, and practices on gender equality related activities in Mediterranean higher engineering education's institutions—members of the Network of Mediterranean Engineering Schools (RMEI). It addresses efforts for an inclusive education and a reflexive transformation of Mediterranean engineering schools through a Change Lab created by the RMEI with the support of the EU HORIZON2020 TARGET project '*Taking a Reflexive Approach to Gender Equality for Institutional Transformation*' (http://www.gendertarget.eu/tag/rmei/).

The objectives of the study and of the gender equality social lab are:

1.  Contributing to filling of the gap of methodological approaches on Social Change Labs that carry research on social changes such as gender equality, for the transformation of universities—members of a network of engineering institutions from almost all Mediterranean countries (not only European-Mediterranean).
2.  Targeting the scholarly and academic awareness towards inclusive higher engineering education of the 21st century.
3.  Proposing an empirical framework focusing on changing the role of women at higher engineering education of Mediterranean institutions.
4.  To show how different building blocks of the Change Lab's environment emerged from a Living Lab on technological innovations within the RMEI network, contribute to the outputs of gender equality interventions and culture change.
5.  To explore the role of a context-based network in contemplating social innovations by reframing activities accordingly.

6.　　Sharing the acquired experience and knowledge by suggesting practical design guidelines on how a network-based Change Lab on gender equality can be designed to interact at the level of community of practice, member-institutions, and the network itself.
7.　　Offering a methodological setup.

　　The innovative aspects of the study are:

1.　　Introducing gender equality in conjunction with activities on SDGs innovations taken by engineers.
2.　　The way that a social change (gender equality) is being proceeded in Mediterranean higher education institutions through the activities of a network of universities. Usually, gender equality activities are taken by institutions themselves. In the non-European Mediterranean countries, the gender equality principle in engineering schools is not yet addressed, while in some European Mediterranean countries most of the engineering schools and universities are now taking the leap, addressing the gender equality principle mainly through the harassment problem. Some European Mediterranean engineering schools are in advancement of gender equality sensitive approaches and plans for institutional transformation, and teaching.
3.　　The interdisciplinary dialogue and communication that is now open between two different scientific disciplines (social studies and engineering education) concerning a social change which is not in the domain of practice of engineers. This study brings the light on the importance for engineering scientists (having non-social science background) to build knowledge on how to design effective actions and interventions on gender equality at the bureaucratic and masculine dominated institutions of engineering education, while social scientists, from the other side, are not dealing with the SDGs technological innovation in their practices.
4.　　This study shows how to build bridges that are needed to understand that the interventions for social change in the context of engineering must go hand in hand with their activities on SDGs technological innovations and global sustainability, subjects which engineers work on and where engineering education is transitioning.

## 2. Methodology

　　Our main goal was to define a framework to guide the collaborative advancement of gender equality innovations for the wicked systemic problem of gender inequality in Mediterranean engineering education institutions. Although it has been a long effort by the EU to solve gender equality bottlenecks in academia by using mainstreaming methods and various tools, in Mediterranean countries there are still many challenges changing the priorities imposed by various perceptions and bias, derived from culture, religion, socio-economic differences, technological advances, political and peace crisis (in some Middle East Mediterranean countries), besides the global challenges beyond climate change and other disruptions in the area.

　　For the creation of the Change Lab, our main hypothesis was that if we can manage to introduce common values of sustainable development and equality in a clear, open, and collaborative way to the members of the network (professors and students), we can also catalyze gender equality practices in the engineering institutions of the Mediterranean community. Thus, we considered the "behavioral spillover" which is the notion that "one's behavior triggers the adoption of other behaviors". From an academic perspective, spillover is interesting because it sheds new light on the process of cultural change [14].

　　We envisioned that the behavioral spillover could work with the principle of gender equality because the inclusion principle is of growing importance in many sectors of engineering practices, including education, research, business, and governance. While the 2030 Agenda for Sustainable Development is envisaging building resilient, equitable, and inclusive societies, sustainable development implies constant evolutionary and adaptive change with gender equality to be the prerequisite to this. Gender equality is a critical goal because its implementation can foster positive cascading effects on the achievement of all SDGs, and it is directly connected to the nexus of education-sustainability. Increasing gender equality

will result in a positive impact on productivity, problem-solving, innovation; all of which are essential outcomes for tackling the great challenges we are facing, from health to food security, from climate change to sustainable communities. Ethics/values as agencies should be integrated in policy planning, natural and social capitals, towards the acceleration of the fundamental changes for a sustainable life.

The purpose of the creation of the Change Lab within RMEI was to develop a 'space' that elaborates gender equality solutions in engineering schools of the Mediterranean, while being supported by a team of social scientists—experts from the TARGET project (Horizon2020 project). Traditionally, disciplines (e.g., social science, engineering) have their own processes to solve specific problems through methodology development. For example, engineers are more familiar with Living Labs, while social scientists are more familiar with the Social Labs. Since we aimed at creating a Social Change Lab within an engineering context to be incorporated in the engineering education toolbox, we identified the contributions from both disciplines. On the one hand, science contributes with technical innovation approaches that contain information to solve the wicked, socio-ecological global challenges that need not only technological knowledge but interdisciplinary approaches, simulation models to find patterns and predictions, although, the information may not explicitly refer to the phenomena under analysis (gender equality). On the other hand, social science contributes to the domain of knowledge, language, ethics, priorities, and all hidden aspects of the problem in an engineering method.

We aimed to open a gender equality dialogue through the Change Lab, specifically aimed at sparking dialogue between stakeholders. The Change Lab within RMEI is not like the traditional research labs that universities operate. A key difference in our methodology is the incorporation of people and stakeholders from different schools from Mediterranean countries with different perceptions, cultures, and level of advancement of the gender equality principle in their country. The challenge here is the lack of homogeneity in interventions and consensus by all member institutions. Thus, a set of crucial principles in the stage of the Lab design considered including participation, inclusion, diversity, commitment to SDGs. As such, several stakeholders are involved with the development and adoption of potential cultural and/or structural change. These stakeholders were already in direct connection with RMEI activities for the Sustainable Development Goals in the Mediterranean territory. Other stakeholders (e.g. from member-institutions) provide access to domain reference data on the gender equality state of the art in their institutions. These are the member institutions' leaders and managers, deans, and rectors of the schools, from whom a commitment is required to build on the institution's transformation. These leaders, although not educated to social science disciplines, are equally relevant as stakeholders, as defined by the participatory framework of the Change Lab. National agencies and women associations are also stakeholders. Students from the member-institutions participating in the Change Lab, educated in the complex-systems analysis, are the change agents.

For a research system in the form of a gender equality Change Lab aiming at having impact on member-intuition's policy and structures, there is a need to be supported in knowledge, methodology, and financially by projects; the factor that the RMEI was missing or it was not always its priority, before the TARGET project. This is enabled by the EU TARGET project (the network is a member of the TARGET project consortium) that aims at gender equality transformation of higher education institutions. Therefore, the methodology is designed to be applied after receiving guidance and support by the EU TARGET project consortium.

In order to bridge the communication between both disciplines (social science and engineering), it is important for engineering scientists (having non-social science's background) to build knowledge on how to design effective actions and interventions on gender equality at bureaucratic and masculine dominated institutions of the Mediterranean engineering education. From the social science domain, the scientists and practitioners of the TARGET project consortium provided information, knowledge, and feedback for the proposed interventions to be adopted by the network. They provided the knowledge for a social change process to be developed by non-social scientists, (engineers) who are the members of the network's Change Lab.

Social scientists of the TARGET project, on the other hand, need to understand that the interventions for social change in the context of engineering must go hand in hand with their activities on SDGs innovations and global sustainability, subjects which the engineers work on and where engineering education is transitioning. For this, gender equality internationally recognized experts were included in the TARGET project consortium, as the advisory committee, to contribute to this bridging and to provide feedback on this innovative approach to a social change.

One of the challenges of generating a general methodology of social labs is the balance between general guidelines and contextual guidelines that attempt to fit local realities and cultural contexts, as it is the case of Mediterranean countries, where important cultural and religious differences exist. The methodology we followed, although engineering-specific, can be applied in other institutions. The design principles are general and can be applied for other contexts. As engineering scientists, we also consulted a methodology that is used in other scientific domains (e.g., in planning and evaluation of urban performance [15]).

Finally, our methodology is grounded empirically in the tailored practices of the gender equality Change Lab, embedded in the network of Mediterranean engineering schools that is a collective organizational integrity, taking into consideration the interconnectedness of SDGs, SDG5 with engineering education and practice. The methodology of this Change Lab is only schematically depicted in Figure 1, in a picture consisting of a domain of interconnections among the network (RMEI), the Mediterranean engineering schools, the TARGET project, the GAMe sub-network of RMEI students (https://rmei.cc/0/index.php/fr/programme/michel-angelo/what-is-game) with SDGs learning in Mediterranean countries higher in engineering education.

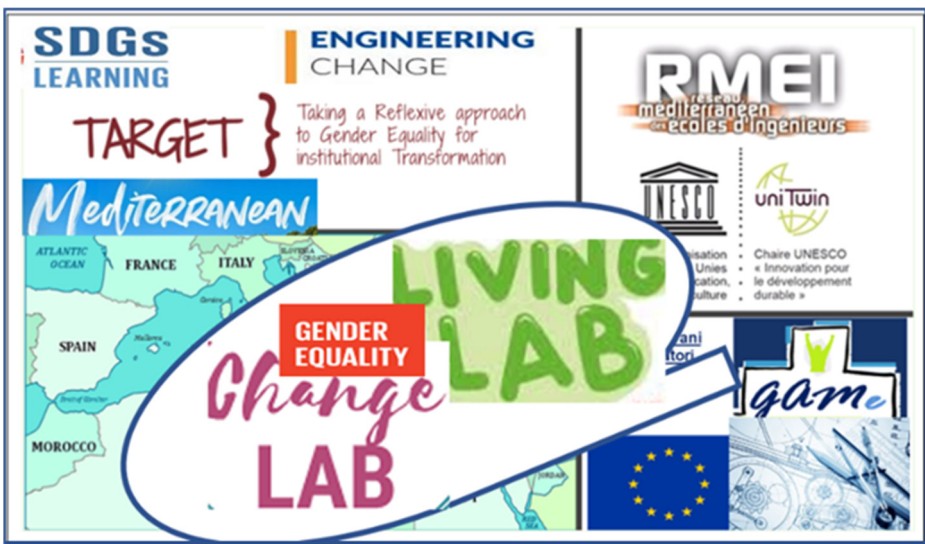

**Figure 1.** Network of Mediterranean Engineering Schools (RMEI's) gender equality Change Lab interconnectedness.

The methodology consists of the following steps:

(1) Design a Change Lab within an existing engineering education network to process both domains (social science and engineering) requirements and some non-traditional sets to reach a first version of interventions.

(2) Taking an action research for a transformative change through the simultaneous process of taking action and doing research, which are linked together by critical reflection. Its output is used to iterate the reflexive transformation to gender equality of higher education institutions [16].

(3) Building knowledge and capacity to congregate the generated knowledge and systems with stakeholders assessing the requirements for the gender equality policy to be adopted in the

member-institutions (real-world). Its output is used to iterate the reflexive transformation to gender equality of higher education institutions.

(4)　Planning interventions and activities in member-institutions along with knowing building activities.

(5)　Taking a reflexive self-evaluation to evaluate the intervention's performance in comparison to a domain reference set empirically to be used for iterating the solution.

These five steps are sequential in appearance; however, the framework could be iterative or nested, depending on the development strategy followed in each institution. Each step receives specific input and tools from both disciplines (social and engineering). All steps use tools from user research methods to enhance communication between people from different backgrounds [17]. This is important, for instance, to ensure that the "language differences" between disciplines converge into opportunities rather than limitations during the development of the process.

We describe each step, in detail, as follows:

### 2.1. Designing the Change Lab

The study is based on empirical evidence of the Network of the Mediterranean Engineering Schools (RMEI) on gender equality. The network (Réseau Méditerranéen des Ecoles d'Ingénieurs-Mediterranean Network of Engineering Schools), was established by ~100 Mediterranean schools of engineering in 1997, in France, with the mission of sustainable development (http://www.rmei.info/index.php/en/). The network is also affiliated to the UNESCO UniTwin chair of sustainable development innovations. Its vision is sustainable development and peace in the Mediterranean, through education on sustainability, responsible research and innovation in engineering, and inclusion.

Thriving for a sustainable world, gender equality is important for the network because it is acknowledged as a social value: Women represent more than half of the world's population and they can contribute to create a sustainable perspective; engineers—men and women—need to work together towards sustainable proposals.

The RMEI network contributes in knowledge and skills-development for engineers to be able to develop creative problem-solving, and technical innovations that are environmentally and socially sound, at a local and global level, by encouraging interdisciplinary thinking of systemic technological, environmental, and social sustainability, responsibility, and gender equality. It was crucial to articulate and share a common purpose within the lab considering more specific design principles.

The network considers learning, training, research and innovation (R&I), and transformation in the context of co-creation. It is a self-organizing system; the relationships within the network are at the core of the most promising capabilities regarding the structural and institutional change. It has as mission to catalyze the transformation of engineering education towards creating an understanding of the complexities of the global and local systems, and helping future engineers, scientists, and managers (men and women) to become able to propose sustainable solutions in their professional life with equality and inclusion, metabolizing all the above into a new mindset.

Concerning the gender equality dimension, the network's mission is offering the support for the generation of a "gender equality culture" for all actors of the collaborative system (professors, students, leaders of the engineering institutions of the Mediterranean countries, etc.). It envisions to contribute to this transformation by (a) taking into account that global challenges are complex and interrelated thus requiring interdisciplinary and system-thinking approaches, and (b) respecting the local cultures.

The network as affiliated to UNESCO UniTwin chair of sustainable development innovations links its efforts to promote the right to education and supports the achievement of the Sustainable Development Goals (SDGs). Through the Education 2030 Framework for Action, SDG5 is devoted to achieving gender equality and empowering all women and girls.

Traditional engineering schools' administration across the Mediterranean world is characterized by governance through hierarchical structures of command and control (in most of the cases by male leaders), including pre-conceived ideas and perceptions on women's position in the school, society, and family.

The network with the TARGET project's support has considered that the design of a Change Lab would be the ideal structure for the reflexive approach for gender equality transformation because it allows the learning and development of gender equality policy, plan, and activities while putting the system of the network's member institutions in the epicenter.

Higher engineering education of the 21st century calls for respect to ecosystems and openness towards society, respecting inclusion and diversity. RMEI acts as a Living Lab of people co-working on SDGs innovations, such as SDG6, SDG7, SDG9, SDG11, SDG12, SDG13, SDG14, SDG15 innovations, directly connected with the engineering R&I and practices. It also acts as a Change Lab of people co-working on SDG5 innovations. In fact, the network supports the responsible research and innovation (RRI) approach which, in practice, takes up the issue of gender and ethics in the research and innovation content and process, formal and informal science education (https://ec.europa.eu/programmes/horizon2020/en/h2020-section/responsible-research-innovation). It also represents contributions from multiple disciplines in engineering fields, systems approach, sustainability, also incorporating social disciplinary backgrounds.

Thus, in the organization chart of the RMEI network the gender equality Change Lab is presented as a sub-lab of the broader Living Lab on sustainable development, created as a facilitating vehicle in this respect.

### 2.2. Action Research

Action research is ideally suited to strengthen Change Lab's ability to meet the dual aim of understanding and promoting emerging social phenomena, such as gender equality in engineering schools. In turn, conducting action research in a Change Lab setting helps to overcome some of the gender equality principle's known bottlenecks [10].

In the systems-theory, action research is an input phase in which we become aware of problems yet unidentified, realize the need to effect changes, and share with experts the process of problem diagnosis. Action research is a cyclical process of change [18]. The cycle here begins with a series of planning actions initiated by the TARGET project. It includes:

(1) Realization and awareness of the problem.
(2) Articulation and sharing of a common purpose.
(3) Diagnosis (gender equality audit), data gathering (survey), feedback of results, and joint action planning.
(4) Transformation process that includes actions related to learning processes, planning and executing changes in the network and member-organizations.
(5) Output/results phase including actual changes in behavior (if any) resulting from corrective action steps taken following the second stage.

### 2.2.1. Realization and Awareness of the Problem

RMEI has had no explicit gender policy in place until 2017. A first attempt to establish a gender working group was made 12 years ago. The leader (female) of the gender equality working group was a very motivated and effective member of the network, a professor and a Rector at the Istanbul Technical University in Turkey. The working group was composed by six professors from engineering schools—members of the network, from Spain, Greece, France, Spain, Turkey, Cyprus. With the strong leadership, the working group had managed to inspire a dialogue on gender equality within the network and catalyze a community of practice and/or plans in some member universities: (a) at the Polytechnic University of Cataluña in Barcelona, Spain (UPC), which started to develop the first plan on gender equality with the support of the Dean of the University who was member of the working group at that time, and (b) the Aristotle University of Thessaloniki in Greece where a center for 'Gender Equality, Space, Technology, and Environment' was created with the support of the Dean of the engineering school. Unfortunately, the breeze for change within RMEI did not last very long due



to the following reasons, according to the author's opinion who was also one of the members of the working group on gender equality at the time:

(a) The founder and leader of the working group who was very much involved in gender equality dialogue, activities, and practices at European level, and thus has had the knowledge, experience, and networking capacity to drive the social change in Mediterranean engineering education, sometime later retired from her university and thus she could no longer represent her university at the RMEI network, (however, she became the president of the European Women Rectors Association, EWORA).

(b) The other members of the network, mainly professors in Mediterranean engineering schools, were somehow lacking in knowledge, experience, and networking at the European level, to effectively drive the social change on gender equality within the network and inspire other members, although they managed to make progress in their universities of origin.

(c) Lack of guidance and methodologies to gender equality cultural and structural change.

(d) Lack of multidisciplinary collaboration with social studies institutions and scholars.

(e) Lack of financial support to organize knowledge and capacity building workshops.

(f) No quantitative data on the sex composition of students and staff of all members were available, nor did a comprehensive understanding of the most relevant problems to be addressed because the engineering schools of the Mediterranean have had no capacity for this kind of data collection.

(g) In addition, cultural, socio-economic, and political disparities across the Mediterranean countries had put barriers in developing an effective line of action to support actual change.

Later on, in 2017, the starting cognitive realization point was the awareness of the extent to which matters related to gender equality are considered by higher engineering education institutions of the Mediterranean. The first motivating point was the willingness expressed by some members of the network to be further involved in a process of gender equality learning and to participate in the developing of the TARGET project proposal. The starting point and anchor of the process was a tailored gender equality plan or strategy in each gender equality innovating institution, which designed, implemented, monitored, in the course of TARGET project.

TARGET is a SwafS-03-2016-2017 Coordination and Support Action funded by the European Commission Program H2020. The main goal of TARGET is to contribute to the advancement of gender equality in research and innovation by addressing gender-related institutional barriers to careers, decision making, and research and innovation, and high education curricula content. The TARGET consortium is complemented by an advisory board composed of seven experts on gender and R&I from transnational networks of women in science. The TARGET approach goes beyond the formal adoption of a gender equality policy by emphasizing an iterative and reflexive process towards equality at the institutional level as well as the establishment of a community of practice for gender equality within the institution. Change is the result of increased institutional willingness and capacity to identify, reflect on, and address gender bias in a sustained way. (https://www.gendertarget.eu/).

2.2.2. Articulation and Sharing of a Common Purpose

Therefore, it was crucial to articulate and share a common purpose within the network, on gender equality. Such shared purposes were key starting points for defining and building the organizational structure (Change Lab) within the network. The starting cognitive shift for a gender equality Change Lab within RMEI was the embedded evidence that:

(a) Research and higher education institutions are gendered settings in the Mediterranean.

(b) Women academics do not rise through the ranks as fast as men do with the same credentials and personal circumstances.

(c) Despite over half of all PhDs being awarded to women, the percentage of female tenured faculty hovers while between 20–33% in the EU, it falls to as low as 5% in fields like engineering, demonstrating the difficulty women face moving up in academia, [19].

### 2.2.3. Diagnosis

One of the first activities of the Change Lab was the launching of a gender equality audit that was the collection of quantitative and qualitative data on gender issues (sex segregated data on students, staff, board of RMEI universities, existence of gender studies, gender policies etc.) in the member-universities. The audit, a survey, served as a baseline analysis of the status quo of gender equality in the member-institutions. Based on that, relevant gender gaps were identified, and the first discussion of gender equality priorities took place.

### 2.2.4. Transformation Process

Transformation refers to a process of inspiring, catalyzing formal and informal cultural changes, and shaping policy, strategy, and plans to generate new meanings and new visions of the future. The transformation process started with the TARGET project entitled "Taking a Reflexive Approach to Gender Equality for Institutional Transformation," aiming at initiating sustainable institutional change in seven gender equality innovating institutions in Mediterranean countries (https://www.gendertarget.eu/).

The first financial and capacity-building support came by the TARGET project (2018–2021), in 2018. TARGET is the tool that enables RMEI to link knowledge with action, enhances collective action, and promotes social learning, and the gender equality principle by creating the gender equality change lab.

With the support of the EU HORIZON2020 TARGET project regarding the specific vision of structural and institutional change towards gender equality, RMEI is cultivating a strong potential regarding institutional change on gender equality by driving the awareness and mobilization of its resources through the social lab, at the level of:

(a)　Network itself (board, working group, policy, strategy) and
(b)　Universities that are members of the network.

### 2.3. Knowledge and Capacity Building

Learning for gender equality refers to the learning experienced by all those engaged in sustainability, including learners themselves (professors and students), facilitators, coordinators, and university leaders [20].

Institutional workshops were organized, open to the members of the Change Lab aiming at a gender equality co-learning. They were focused on the development of knowledge, capacity, and competences. Learning-based case studies were shared.

Structures for gender equality (committees or centers) in some member-institutions were created with the support of the Change Lab. They are targeting more specific measures on gender equality within the institution (recruitment and empowerment of gender equality officers and academics, promotion, etc.).

### 2.4. Planning

A gender equality policy statement was developed and discussed in many meetings of the members of the Lab and presented for an open dialogue during the network's general assembly where the policy statement was approved unanimously. Once the dialogue allowed agreement on a policy statement, the output was to plan activities that had the potential to open the problem up to a dialogue on the member institutions, and national stakeholders.

National workshops were co-created by the change Lab and the member-institutions and brought attention to local stakeholders. For the national workshops, a stakeholder approach was adopted. Key priority issues to promote gender equality and counteract gender bias were identified prior to the workshop, put on the consultation agenda, and raised during the consultation process, always paying attention to the context particularities.

*2.5. Reflexive Evaluation*

The approaches and tools were developed from the perspective of the engineering schools (applicability, practicability), in terms of consolidating a sustainable line of action to support institutional change in member universities. The feedback to the above has especially considered the cultural differences between countries. For the self-evaluation we used context and performance monitoring and evaluation indicators, to fit the specificities of the Mediterranean countries' engineering education.

Based on the TARGET project's evaluation framework and guidelines, and on members' creativity, a tailored, bottom-up, and case-specific evaluation process traced the interventions empirically. Tailored indicators to evaluate the gender equality progress, considering the systemic view of the cross-cutting gender analyses interdependently connected to SDGs innovations, were applied.

## 3. Design Principles

We present in this section, first, the design principles of the Living Lab, which corresponds to the whole network in learning for sustainable development innovations, and secondly, the design principles of the Change Lab, which is a sub-lab of the Living Lab. We present both in order to distinguish their differences.

*3.1. Design Principles of the RMEI Living Lab on Sustainable Development*

The main design principles for the RMEI Living Lab on sustainable development are presented and analyzed in Table 1.

**Table 1.** Main design principles of the Living Lab on sustainable development (RMEI network).

| No | Principle | Description |
|:---:|---|---|
| 1 | Vision and commitment | Professors and students feeling committed to SDGs implementation are the persons who embrace the Living Lab. They are the visionary, collaborative people that trust the network and embrace its vision. |
| 2 | Transdisciplinary agenda | Engineering and technology disciplines related to SDGs (SDG6—clean water and sanitation for all; SDG7—access to affordable, reliable sustainable energy for all; SDG9—strong and resilient infrastructures; SDG11—inclusive, safe, resilient, and sustainable cities; SDG9—inclusive and sustainable production and industrialization; SDG13—climate action). |
| 3 | Transdisciplinary partnerships | It is acknowledged that sustainable development has an ecological, socio-cultural, economic, and spatial dimension. |
| 4 | Interdisciplinary working groups | The working groups can submit ideas/proposals for funding to outreach sources. For each idea, the project leader who usually is member of the management board of the network invites colleagues from other members institutions within the network to become involved in the specific initiatives and collaborate with other stakeholders. |
| 5 | Open participation | Provide a space for everyone in the network to be engaged in activities, strategies, and projects for sustainable development. Based on the knowledge exchange, the network can develop further needs-oriented development strategies. |
| 6 | Working group | A multinational, inter-generational, inter-gender, multicultural, multidisciplinary working group of 12 persons was created |

Schematically, Figure 2 presents the nexus of RMEI network facts and forces on innovation, Living Lab's approach and its environment. RMEI has followed an approach that was considered appropriate for the: (a) Design of an innovation Living Lab based on principles that integrate and correspond to the network's vision, member institutions leaders' commitment, existing agenda of scientific areas, partnership and collaboration under a working group, and (b) assessment of the impacts related to

the increase of collaborations within the network, of synergies with other SDGs innovations, number of projects addressing the sustainable development of the Mediterranean and the boosting of ethical value for peace in the region, the creation of trust and friendships.

In Figure 2, the Living Lab's environment integrates the frameworks of sustainability, systems-thinking, complexity transdisciplinarity, interdisciplinarity, and ethical values that are the cornerstones of engineering education of the 21st century. The RMEI's Living Lab, as Figure 2 shows, is an engineering-context, Mediterranean-centered lab that uses human resources that the member-institutions provide, and the students of the member-engineering schools to advance the education on global challenges and prepare engineers to work on sustainable innovations.

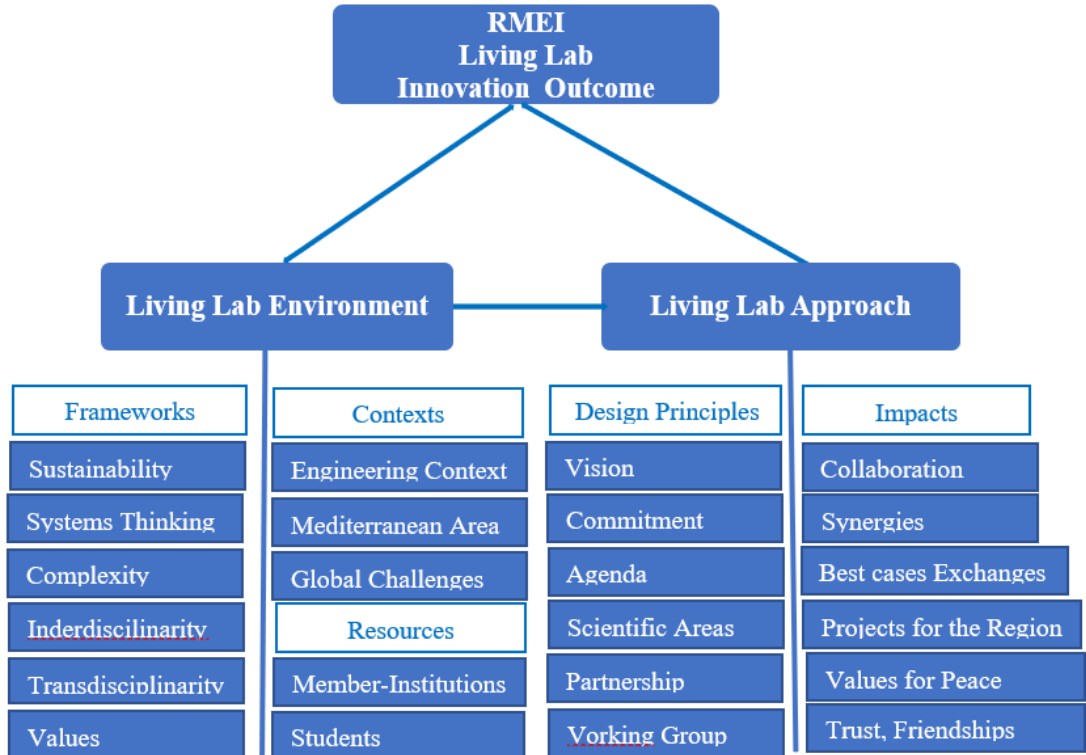

**Figure 2.** RMEI's Living Lab for sustainable development innovations.

*3.2. Design Principle of the RMEI Change Lab on Gender Equality*

The specific principles followed for the design of the Change Lab within RMEI are shown in Table 2.

**Table 2.** Design principles for the Social Change Lab.

| No | Principle | Description |
|----|-----------|-------------|
| 1 | Specific objectives | • Awareness raising. <br> • Space for experiential learning and feedback from professors, students. <br> • Knowledge exchange between member institutions. <br> • Space for dialogue. |
| 2 | Focused targets | Focus on gender equality capacity building within the network and across member institutions. |



**Table 2.** *Cont.*

| No | Principle | Description |
|---|---|---|
| 3 | Stakeholders engagement | Including related associations of the countries and management authorities engaged in schools' management and development. |
| 4 | Capacity building | For gender equality in member engineering schools that are not yet reflecting on the processes. |
| 5 | Aligning agendas | It does not equate to sharing one specific case from institutions that are successful and running much in front on gender equality plans compared with the others, but inviting a variety of cases that can all contribute to a common purpose. |
| 6 | Creativity | Finding creative ways to inspire a cultural change for gender equality in higher education institutions by using art, painting, theatre, poetry, narratives, role-models, and interviews of scholars. |
| 7 | Communicating | <ul><li>The core values.</li><li>Of what the core of the problem is.</li><li>The values that should guide the process.</li><li>What the outcome of the change should be in the society and economy.</li></ul> |
| 10 | SDGs agenda | Working synergistically with the commitment to SDGs agenda. |
| 12 | Dialogue platforms | This dialogue can create innovation in the actions. |
| 13 | Accepting contradictory concepts and ideologies | Contradictory concepts and ideologies can meet in the Change Lab which inevitably involves some level of compromise. |
| 14 | Members' personal development | Open to dialogue, respect the different opinions, understand the differences, and trust the network and the diversity of perceptions on gender equality in academia. |
| 14 | Students' cognitive development | At some point of their development, students can gain such an inspiration within the Change Lab, driving their interests in life and the pursuit of their post graduate studies (e.g., towards Master and PhD studies on the topic). |
| 15 | Reflexive approach | Focused on an open ongoing dialogue with feedback loops. |

Figure 3 schematically depicts the nexus of RMEI network facts and forces, Social Change Lab's approach and its environment. RMEI has followed an approach that was considered appropriate for the: (a) Design of a Social Change Lab within the Living Lab, based on principles that integrate and correspond to the TARGET project approaches, methodologies, and reflexive assessment/monitoring of the progress, new agenda of areas for action, and dialogue on gender equality and communication; (b) assessment of the impact on the cultural and structural changes within the network and member institutions, changes on the management of the network, creation of gender equality policy within RMEI, building of a community of practice in the Mediterranean, and creation of centers/committees of the gender equality issue in member institutions.

In Figure 3, the Social Lab's environment integrates the frameworks of SDG5 on the empowerment of women in academia, fairness and inclusion in Mediterranean higher engineering education towards the economic growth of the Mediterranean region that faces many socio-economic and environmental challenges. The RMEI's Change Lab, as Figure 3 shows, is an engineering-context, Mediterranean-centered lab that catalyzes the gender equality changes with the support of the

(a) Human resources the member institutions provide.
(b) Students of the member engineering schools.
(c) TARGET project knowledge and financial resources.

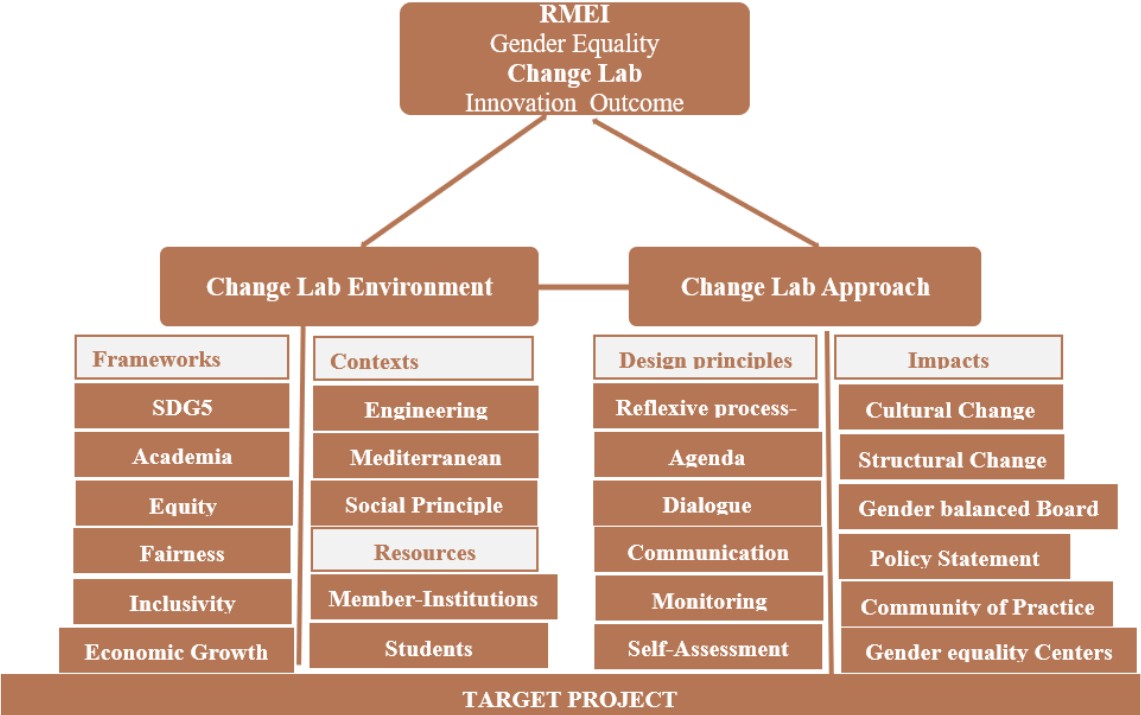

**Figure 3.** RMEI's Change Lab for gender equality in Mediterranean higher engineering education and research.

## 4. Outputs/Results

Social learning according to researchers is defined as "bringing together people from various backgrounds with different values, perspectives, knowledge and experiences to creatively find answers to questions lacking ready-made solutions" [21]. The RMEI Change Lab became a space that offers to practitioners (engineers who do not know the theory of change, but they act for this change) the instruments for analyzing bottlenecks in their work practices in academia; it allows for changing culture, develops tools and puts them on trial. It has also become a forum for the cooperation between experts and local practitioners and relies on the physical attendance of the community in dedicated physical spaces (France and other Mediterranean countries). All the above agree with the definition of a Change Lab given some years ago by researchers on education and pedagogy [22].

The Change Lab is helping the network and the community to discuss the gender equality problems faced in the academic real world, systematically analyze the systemic causes of these problems, design and implement new activities to alleviate the root cause of the problems. Finally, it helps in creating a new mindset and culture by exploring linkages between engineering-specific cultural narratives and gender-equality planning through the lens of the '*culture*' concept.

It delivers a transdisciplinary initiative (TARGET project) and applies innovative approaches for scaled and systemic transformation processes at the Mediterranean engineering schools (members of the network), by actionable processes (workshops and projects). It also creates an understanding of the complexities of socio-ecological systems and helps future engineers, scientists, and managers (men and women) to build sustainable and resilient societies.

Besides innovative contents and practices specifically related to a reflexive transformation for gender equality, the Change Lab takes a more holistic integration of SDGs by leading awareness, fostering cultural shifts, building capacity, and developing a community of practices, in different member universities of the North and South Mediterranean basin. By gathering different cultural entities together and around common values and vision, it encourages member organizations to

challenge and question their informal norms of gender eqyality. These norms are differentiated by important cultural, political, economic factors of each member.

It creates a multiplier effect at member institutions and brings formal and informal cultural changes. Change Lab's interventions have an impact on students participating in the network's activities (the sub-network called GAMe), on networks' board (a gendered-balanced board achieved), and on some member-institutions' re-organization (gender equality committees' creation), while the role of the management is key in sustaining the change effort.

The process of participation in the Change Lab has an impact on the collective transformative agencies (professors, students, leaders), by reallocating the focus of discourse from blaming systems, to committing themselves to a shift in thinking (individual action) to implement coordinated actions (collective actions). Several volunteer-students expressed their willingness to co-organize a workshop and to carry out insightful presentations on gender equality topics in conjunction with topics on other SDGs, preparing videos, interviewing keypersons, etc. Some of the students are now very sensitive to the equality principle and inclusivity matters and wish to follow post-graduate studies on gender equality topics, always in conjunction with sustainability, resilience, and leadership concepts. Therefore, the Change Lab creates value for the network and the community [23].

Finally, the Change Lab is envisioning to have an impact on the operational practices of Mediterranean engineering schools, and later to integrate the gender equality dimensions into the teaching curricula as well.

The profile of the Change Lab is depicted below:

*Profile of the RMEI Change Lab*

Based on a focused foresight process, an inter-sectoral profile of the RMEI Change Lab was elaborated with the following characteristics:

(a) Hybrid

It is a hybrid niche positioned at the boundary between network self-assembling structures and member-institutions.

(b) Transparent Leadership

It has transparent leadership and organizational structures, tailored to sustainable development specific goals, common values distributed transparently.

(c) Good Communication–Collaboration

It has good communication and fair collaboration with member institutions.

(d) Takes a Long-Term Change

It aims to create a long-term change in the network's culture and to maximize learning by member universities.

(e) Creates Active Learners

Students are becoming 'active learners' because they are far from the realm of structural institutions (universities) and rigid teacher–student roles.

(f) Creates Change Agents

Participants can act as 'change agents,' advancing their own education experience, participating in the design of a gender equality plan, targeting to increase influence in the community of Mediterranean engineering institutions, by sharing their learning with others and by exchanging good practices.

(g)   Creates Synergies

The key components of the Change Lab constitute important synergies with other SDGs innovations that enhance the process and the equality and fairness principles.

(h)   Practitioner Character

It has a practitioner's character because engineers who are participating in the Change Lab are not acquainted with social change scientific methodologies but rather with technological innovations. Therefore, from a practitioner's perspective, the notion of behavioral spillover was attractive because of the promise of changing behaviors in a cost-effective manner and might bring little institutional change and regulation which usually is difficult.

(i)   Multi-stakeholders Participation Character

Multiple-stakeholders participation was used in discussing gender equality via the co-organized national workshops. Data collected by audits were used to get a rich picture of what is happening on the ground. However, knowledge and information will remain an insufficient and uncertain basis for guiding our path into the future if there is no transformative change in the way of our thinking and acting.

(j)   Fits Agendas

To make it methodologically easier, participants from varying specific engineering-scientific backgrounds and levels in the academic pyramid are looking for different ways that may help advance their school's interests in the developed gender equality committee/center. Therefore, the agendas of the various committees/centers on gender equality developed in different schools were different, fitting in the institution's ability, willingness, and conceptions, sometimes at the expense of the Change Lab's proposed agenda. So, the lenses used to interpret the gender equality necessity and its advancement processes at the institutional level, in different engineering schools, are different and tied to their national culture. Therefore, the emphasis of the Change Lab is put on the inspiring and encouraging change process, and monitoring and evaluation process that emphasizes critical reflection and reciprocal learning, not with the purpose of ranking schools or practices, but rather to stimulate learning and action, taking lessons from others.

(k)   Case-Studies and Best Practices Exchanger

Case-studies of gender equality on institutional, national, Mediterranean, and international levels looked at co-learning and co-exchanging the best processes currently used in the more advanced in gender equality engineering institutions of the Mediterranean. This is the case of European Mediterranean countries' universities that have national laws on gender equality, compared to the Middle East countries and North African countries.

(l)   Contributes to Cultural Narratives

To understand a human being and actions, thoughts, and reflections, we must look at the social, cultural, and institutional context in which they operate. According to Vygotsky (1978) [24], human learning and development occur in socially- and culturally-shaped contexts, meaning that people become what they are depending on what they have experienced in the social contexts in which they have participated. To fully understand the problem of persisting gender inequality in academia, to design plans and institutional change processes, scholars and practitioners often focus on cultural narratives. Cultural narratives serve in different ways as support factors for gender equality activities planning and implementation if they are mindful [25]. Cultural narratives, however, as support factors require strategic communication, leadership commitment, and reflexive self-evaluation from one side and to give sense, direction of anticipations to participants and practitioners from the other side [26]. The Change Lab

during the annual MICHELANGELO workshop that the RMEI students organize every year, with the leadership of a professor who is also member of the Change Lab, uses narratives encompassing a range of approaches, including ancient Greek-Roman theater, Mediterranean poetry, Mediterranean cultural representations, Ancient Greek-Roman-Islamic, and in general Mediterranean women's biography, personal narratives, history, and journalism.

(m)　Creates New Forms of Learning in an Ever-Changing World

Finally, introducing new forms of learning in an ever-changing world is required for institutional transformation. It is commonly recognized that transforming an activity needs a form of collaboration that crosses established organizational boundaries [25]. When a group of people searches collaboratively how to change a culture and form new activities in which they are engaged, we could speak of shared transformative agency [25]. Cooperation among several higher educational institutions and other non-academic partners can create networks that are based upon a common will to share and extend their experiences beyond the context of their initial community. Networks can link knowledge with action, enhance collective action, promote social learning [27]. Trustworthy networks can enhance the willingness of people and institutions to act [28], by utilizing the network's human and financial resources and also governance structures of member institutions that are located in different countries, for the accomplishment of common goals [29], realizing the Living and Change Lab concept. Networks can play a key role in contributing to social transformation and sustainability-oriented engineering education if sustainable development is the epicenter of their mission and the contemplation of social change.

## 5. Conclusions

This study brings an empirically grounded case study of a Change Lab and contextualizes the above with practical design principles. It also illustrates the innovative potential of the Change Lab of the RMEI engineering network to foster social learning, create value for participants and their community (engineering community).

It is a practice-driven creation, representing a pragmatic approach to SDG5 innovations. It is characterized by a reflexive approach of interventions in real-life, with the active involvement of users (member-institutions), and the support (financial and experience-transfer) of the EU TARGET project. It has the potential to be an instrument for the active inclusion of members in activities, investigating socio-spatial questions on the socio-cultural dimensions of gender equality application in academia.

The RMEI Change Lab benefits from the existing co-sharing and established collaboration conditions within the network's members. It generates a cultural and institutional structural change in the context of engineering schools in Mediterranean countries, by involving a range of stakeholders (policy-makers, practitioners, administrators, researchers, etc.).

We advocate that collaborative learning in networks and in social change labs is challenging the melting of traditional boundaries around engineering institutions, by moving from separate bureaucratic centrally-managed institutions to interconnected ecologies of self-assembling collaboration with mutually beneficial partnerships, where individual learners are becoming change agents to kick off the output phase of the other learners' metabolism, and finally create an ecosystem of metabolizing learners.

**Funding:** Funding received from the European Union's Horizon 2020 research and innovation program under grant agreement No 741672.

**Acknowledgments:** The author would like to thank all members of the RMEI Change Lab: O. Boiron, executive director of RMEI office, ECM-France, Fatma Asfour, vice-president of RMEI, University of Cairo, Egypt; Massimo Guarascio (leader of GAMe), Sapienza University, Italy, Najoua Essoukri Ben Amara and Ghiss Moncef at ENISo-Tunisia; Khalid Najib and Ibtissam Medarhri, ENSMR-Maroc; Tilda Karkour Akiki, Associate Dean of School of Engineering, Holy Spirit University of Kaslik, Lebanon; Khaled Al-Sahili, Dean of the Engineering School, at Al-Najah National University, Nablus, West Bank, Palestine; Juan Jesus Perez, Vice Dean at the Universita Politècnica de Catalunya (UPC), Barcelona, Spain and all students from RMEI-GAMe are acknowledged. Authorities of the member-institutions and various stakeholders participated in the workshops are acknowledged.

**Conflicts of Interest:** The author declares no conflict of interest.

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
