# Peer review of "New Forms of Social Learning in Mediterranean Higher Engineering Education: Change Lab for Gender Equality Transformation, Methodology, Design Principles"

_sustainability, doi:10.3390/su12166618_

Round 1

Reviewer 1 Report

The paper is proposing an empirical framework focusing on gender equality at higher engineering education of Mediterranean institutions thru the change lab embedded in the RMEI. It seems very interesting work and I see detailed design principles and plans. I see this paper resubmit to the journal and the authors reorganized the paper. However as a reviewer, I still do not see much novelty of research in sustainability at all. 

While I am reviewing this paper, I feel like this paper is a proposal discussing potential of the Change Lab. I do not see clear objectives of the study or goal. There is no results other than proposed plans only.

The author said RMEI’s attempt to eatable a gender working group was not successful and listed a few reasons. Are there any results of evidence for the reasons? Why does the author think it was not successful? Any reference or report addressing the failure?

There are some figures and tables but the author did not discuss them much but introduced them only.

This link does not work

https://www.rmei.info/index.php/fr/programme/michel-angelo/what-is-game

So many citation from [14] which is the author’s other paper.

The title is still so long.

I have seen a lot of grammar problem thru the entire paper and the author must revise it carefully.

The article type of this paper must be a concept report or others other than 'article'. 

Author Response

REPLY TO REVIEWER 1 COMMENTS

Comment 1.      The paper is proposing an empirical framework focusing on gender equality at higher engineering education of Mediterranean institutions thru the change lab embedded in the RMEI. It seems very interesting work and I see detailed design principles and plans. I see this paper resubmit to the journal and the authors reorganized the paper. However, as a reviewer, I still do not see much novelty of research in sustainability at all. While I am reviewing this paper, I feel like this paper is a proposal discussing the potential of the Change Lab. I do not see the clear objectives of the study or goal. There are no results other than the proposed plans only.

Reply :                The resubmitted manuscript has completely restructured. However, Thank you for the suggestion to highlight the aim and objectives. A new sub-paragraph has added, to highlight the aim, objectives and innovative aspects, the following:

                            1.1 Aim, Objectives, and Innovative Aspects of the Study

                            This paper aims to introduce the dialogue and practices on gender equality related activities in the Mediterranean Higher Engineering Education’s Institutions-members of the RMEI network. It addresses efforts for inclusive education and a reflexive transformation of Mediterranean Engineering Schools through a Change Lab created by the Network of Mediterranean Engineering Schools (RMEI) with the support of the EU HORIZON2020 TARGET project ‘Taking a Reflexive Approach to Gender Equality for Institutional Transformation”( http://www.gendertarget.eu/tag/rmei/).

The objectives of the study and of the social lab are:

  • Contributing to the filling of the gap of methodological approaches on Social Change Labs that carry research on social changes such as the gender equality, for the transformation of Universities- members of a network of Engineering Institutions from almost all Mediterranean countries (not only European-Mediterranean).
  • Targeting scholarly and academic awareness towards an inclusive Higher Engineering Education of the 21st
  • Proposing an empirical framework focusing on changing the role of women at Higher Engineering Education of Mediterranean Institutions.
  • Showing how the different building blocks of a Change Lab’s environment emerged from a Living Lab on SDGs technological innovations within a network, contribute to the outputs of gender equality interventions and culture change.
  • Exploring the role of a context-based network in contemplating social innovations by reframing activities accordingly.
  • Sharing the acquired experience and knowledge by suggesting practical design guidelines on how a network-based Change Lab of gender equality can be designed to interact at the level of community of practice, member-institutions and the network itself.
  • Offering a methodological setup.

The innovative aspects of the study are:

  1. Introducing gender equality in conjunction with activities on SDGs innovations taken by engineers.
  2. The way that a social change (gender equality) is being proceeded in the Mediterranean Higher Education Institutions through the activities of a network of Universities. Usually, gender equality activities are taken by institutions themselves if they take ones. In the non-European Mediterranean countries, the gender equality principle in the Engineering Schools is not yet addressed while in some European Mediterranean countries most of the Engineering Schools and Universities are now taking the leap, mainly addressing the gender equality principle through the harassment problem. Some European Mediterranean Engineering Schools are in the advancement of gender equality sensitive approaches and plans for institutional transformation and teaching of the inclusion and equality principles.
  3. The interdisciplinary dialogue and communication between two different disciplines (social studies and engineering education) for a social change which is not the domain of practise of engineering education. This study brings the light on the importance for engineering scientists (having non-social science background) to build knowledge on how to design effective actions and interventions on gender equality in bureaucratic and masculine dominated institutions of the engineering education. Social scientists from the other side are not dealing with SDGs technological innovation in their practices.
  4. This study shows how to build bridges that are needed to understand that the interventions for social change in the context of engineering must go hand in hand with their activities on SDGs technological innovations and global sustainability, subjects where the engineers work on and the engineering education is transitioning.

Comment 2.            The author said RMEI’s attempt to eatable a gender working group was not successful and listed a few reasons. Are there any results of evidence for the reasons? Why does the author think it was not successful? Any reference or report addressing the failure?

Reply:                      Explanations are given in the text. It was due to lack of financial and knowledge support.      

Comment 3.            There are some figures and tables, but the author did not discuss them much but introduced them only.

Reply: Tables and figures are self-explanatory. We preferred to structure some parts in the form of Tables for diversity in form.

Comment 4.            This link does not work

                       https://www.rmei.info/index.php/fr/programme/michel-angelo/what-is-game

Reply:                       This link was replaced by the link https://rmei.cc/0/index.php/fr/programme/michel-angelo/what-is-game

Comment 5.            So many citation from [14] which is the author’s other paper.

Reply:                       Ref 14 has cited only once in the new resubmitted manuscript.

Comment 6.               The title is still so long.

Reply:                         The title has been shortened

Comment 7.               I have seen a lot of grammar problem thru the entire paper and the author must revise it carefully.

Reply:                         They are corrected as much as possible.          

Comment 8.               The article type of this paper must be a concept report or others other than 'article'. 

Reply:                         Great, we will ask the editor to change it to a concept report.

Thank you very much

Reviewer 2 Report

The article is grounded empirically in the tailored practices of the gender equality Change Lab embedded in the network of Mediterranean Engineering Schools, a collective organizational integrity. This work highlight the diversity and gender equality and, in this context, attention is paid to the role of TARGET project. The work uses, as methodology, the action research and practical learning, but this part can be more clarified.

But, in section 2. Methodology, you must clarify the methodology of the paper of that which is the methodology of the Change Lab.

If possible, in the table 2. improve the description of: "6 Creativity - Finding creative ways to inspire".

You can improve the section "5. Conclusions", in order to clarify the main objective of the article, as a scientific paper. 

The bibliography seems to be current, sufficient and adequate.

Author Response

REPLY TO REVIEWER 2 COMMENTS

Comment 1.      The article is grounded empirically in the tailored practices of the gender equality Change Lab embedded in the network of Mediterranean Engineering Schools, collective organizational integrity. This work highlights the diversity and gender equality and, in this context, attention is paid to the role of the TARGET project. The work uses, as methodology, action research and practical learning, but this part can be more clarified.

                            But, in section 2. Methodology, you must clarify the methodology of the paper of that which is the methodology of the Change Lab.

Reply:                 The methodology has very much clarified in paragraph 2. Actually the whole manuscript has restricted and clarified.

Comment 2.       If possible, in table 2. improve the description of: "6 Creativity - Finding creative ways to inspire".

Reply:                 Thank you for the comment

                            We added the following:

                            ‘Finding creative ways to inspire a cultural change for gender equality in Higher Education Institutions by using art, painting, theatre, poetry, narratives, role-models and interviews of scholars’.

Comment 3.      You can improve the section "5. Conclusions", in order to clarify the main objective of the article, as a scientific paper. 

Reply:                 Thank you for your constructive comment. E clarified the main objectives in the new sub-paragraph 1.1: Aim, objectives and Innovative aspects, and a new paragraph 5: Results is better elaborated to present the results

                           The first reviewer suggests changing the article from ‘research article’ to concept article”. Do you agree?

Comment 4.      The bibliography seems to be current, sufficient and adequate.

Reply:                 Thank you

Thank you very much

Round 2

Reviewer 1 Report

The authors address most of comments appropriately. However still the authors need to clarify the following comment again.

"Comment 2. The author said RMEI’s attempt to eatable a gender working group was
not successful and listed a few reasons. Are there any results of evidence
for the reasons? Why does the author think it was not successful? Any
reference or report addressing the failure?
Reply: Explanations are given in the text. It was due to lack of financial and
knowledge support."

We don't know the response is the authors' opinion or it is objective data. I asked to show reference (i.e., report or internal data to prove it)?

"Comment 3. There are some figures and tables, but the author did not discuss them
much but introduced them only.
Reply: The Tables and figures are self-explanatory. We preferred to structure
some parts in the form of Tables for diversity in form."

This response is not appropriate at all. You preferred to structure like this but readers including me may not.

For example, in Figure 2, I don't understand much about the schematic diagram that you provided with. There is an tiny arrow from Living Lab Approach to RMEI in one way while I see two way arrow between Living Lab Environment and RMEI. What does this mean? I guess both must be a two-way arrow. I see Figure 3 with two-way arrow. Then The line between Environment and Approach means what?

The items list below are frameworks, contexts, design principles, impacts and resources. Are this under RMEI or some under Environment or Approach? What are the middle vertical lines?

Table 1 describes 5 principles and all are listed in Figure 2 while there are two new principles are added in Figure 2 - Agenda and Scientific areas. 

Similar issue to Table 2 and Figure 3.

what is the Target Project? What do you want to tell?

Figure 3 title is Gender equality Change Lab of RMEI? or Structure of ....

or Schematic Diagram of ....

Figure 2 has same issue with the title.

The diagrams are poorly made. 

What is vorking Group and Inderdisciplinarity?  What is Reflexive process-?The authors seem too hurry to submit the revision. These can be detected in your word system. In my PDF, it shows the red line which can be fixed. The authors said fixed as much as - But not a big deal.

The bottom line is the Tables and figures are NOT self-explanatory. 

Without changing, the article type of the paper, this paper can't be published under ARTICLE.

The comment left above is minor even I wrote a lot. 

In the future, the authors may observe the dialogue and practices then compare with the former system or other systems..then try to submit under research article.

Author Response

REPLY TO REVIEWER 2 COMMENTS

Comment 1. The authors address most of comments appropriately. However still the authors need to clarify the following comment again.

REPLY 1: Thank you very much for your very detailed work on our manuscript

Comment 2. The author said RMEI’s attempt to eatable a gender working group was not successful and listed a few reasons. Are there any results of evidence for the reasons? Why does the author think it was not successful? Any reference or report addressing the failure?

REPLY 2: I upgraded  the explanation by adding the following in the paragraph 2.2.1. The response is the authors' opinion who was a member of the working group of the time and is a member of the board of RMEI.

‘RMEI has had no explicit gender policy in place until 2017. A first attempt to establish a gender working group was made 12 years ago. The leader (female) of the working group was a very motivated and effective member of the network, a professor and Rector at the Istanbul Technical University in Turkey. The working group composed by 6 professors at engineering schools -members of the network from Spain, Greece, France, Spain, Turkey, Cyprus. With this strong leadership the working group managed to inspire a dialogue on gender equality within the network and to catalyze a community of practice or/and plans in some member-universities, as it was the case of Technical University of Catalunya in Barcelona in Spain that started to develop the first plan on gender equality with the support of the Dean of the University who was member of the working group and the Aristotle University of Thessaloniki in Greece where a center for ‘Gender Equality, Space, Technology and Environment’ was created with the support of the Dean of the engineering school.  Unfortunately, this force for change within RMEI did not last very long due to the following reasons, according to the author’s opinion and perception who was also one of the members of the working group on gender equality at the time:  

  1. The founder and leader of the working group who was very much involved in gender equality dialogue, activities and practices at European level, had the knowledge, experience, and networking capacity to drive the social change in Mediterranean engineering education, sometime later retired from her University and thus she could not anymore represent her University in the RMEI network (However, she became the president of the European Women Rectors Association-EWORA).
  2. The other members of the network, professors in Mediterranean engineering schools were somehow lacking behind in knowledge, experience and networking at European level to drive effectively the social change on gender equality within the network an inspire other members, although they managed to make a progress in their universities of origin.
  3. Lack of guidance and methodologies to gender equality cultural and structural change.
  4. Lack of multidisciplinary collaboration with social studies institutions and scholars.
  5. Lack of financial support to organize knowledge and capacity building workshops.
  6. No quantitative data on the sex composition of students and staff of all members were available, nor a comprehensive understanding of the most relevant problems to be addressed, because the engineering schools of the Mediterranean had no capacity for this kind of data collection.
  7. In addition, cultural, socio-economic, and political disparities across the Mediterranean countries had put barriers in developing an effective line of action to support actual change. ‘

Comment 3. There are some figures and tables, but the author did not discuss them much but introduced them only. This response is not appropriate at all. You preferred to structure like this but readers including me may not. For example, in Figure 2, I don't understand much about the schematic diagram that you provided with. There is an tiny arrow from Living Lab Approach to RMEI in one way while I see two way arrow between Living Lab Environment and RMEI. What does this mean? I guess both must be a two-way arrow.

REPLY 3. The following sentence is added (lines 444-457) to explain the figure 2 as requested. The items list below are frameworks, contexts, design principles, impacts and resources under RMEI and with some information of other living labs approaches.  

‘, Figure 2 presents the nexus of RMEI network facts and forces on innovation, living lab’s approach and its environment created. RMEI has followed an approach that considered appropriate for the: a) design of a innovation living lab based on principles that integrate and correspond to the network’s vision, member-institutions leaders’ commitment, existing agenda of scientific areas, partnership and collaboration under a working group, and b) assessment of the impacts related to the increase of collaborations within the network, of synergies with other SDGs innovations, number of projects addressing the sustainable development of the Mediterranean and the boosting of ethical the value for peace in the region, the creation of trust and friendships. In Figure 2, the living lab’s environment integrates the frameworks of sustainability, systems thinking, complexity transdisciplinarity, inderdisciplinarity, and ethical values that are the cornerstones of the engineering education of the 21st century. The RMEI’s living lab, as Figure 2 shows is engineering-contexed, Mediterranean-contexed lab that uses the human resources that the member-institutions provide and the students of the member-engineering schools to advance the education on global challenges and to prepare engineers to work on their sustainable innovations.’

Comment 4.  I see Figure 3 with two-way arrow. Then The line between Environment and Approach means what? What are the middle vertical lines?

REPLY 4. The following explanatory sentence is added (lines 465-480) as requested:

Figure 3 depicts schematically the nexus of RMEI network facts and forces, social change lab’s approach and its environment created. RMEI has followed an approach that was considered appropriate for the: a) design of a social change lab within the living lab, based on principles that integrate and correspond to the TARGET project approaches, methodologies and reflexive assessment/monitoring of the progress, new agenda of areas for action and dialogue on gender equality and communication,  b) assessment of the impact on the cultural and structural changes within the network and member-institutions, changes on the management of the network, creation of gender equality policy within RMEI, building of a community of practice in Mediterranean and creation of centers/committees of the gender equality issue in member-institutions. In Figure 3, the social lab’s environment integrates the frameworks of SDG5 on empowerment of women in Academia, fairness and inclusivity in Mediterranean Higher Engineering Education towards the economic growth of the Mediterranean region that faces many socio-economic and environmental challenges. The RMEI’s change lab, as Figure 3 shows, is an engineering-contexed, Mediterranean-contexed lab that catalyzes the gender equality changes with the support a) of the human resources the member-institutions provide, b) of the students of the member-engineering schools and c) the TARGET project knowledge and financial resources.

Comment 5. Table 1 describes 5 principles and all are listed in Figure 2 while there are two new principles are added in Figure 2 - Agenda and Scientific areas. Similar issue to Table 2 and Figure 3.

REPLY; Table 1 analyses the 6 principles while Figure 2 depicts schematically them in the whole nexus. I added a sixth line in TABLE 1 as you can see:

6

Working group

A multinational, multigeneration, multigender, multicultural, multidisciplinary working group of 12 persons was created

Comment 6. what is the Target Project? What do you want to tell? What is vorking Group and Inderdisciplinarity?  What is Reflexive process-? The authors seem too hurry to submit the revision. These can be detected in your word system. In my PDF, it shows the red line which can be fixed. The authors said fixed as much as - But not a big deal.

REPLY: TARGET project is shortly descripted in 2.2.1 and the website is provided for further information. Interdisciplinarity and reflexive approach are described also.

Comment 7. The bottom line is the Tables and figures are NOT self-explanatory. 

REPLY:  I added the explanation in the text

Comment 8. Without changing, the article type of the paper, this paper can't be published under ARTICLE.

REPLY: The type has changed.

Comment 9. The comment left above is minor even I wrote a lot. In the future, the authors may observe the dialogue and practices then compare with the former system or other systems..then try to submit under research article.

 REPLY: I am grateful to you for your deep review, time, energy and wiliness to support my advancement in presenting interdisciplinary studies.
